# Sirt1-PPARS Cross-Talk in Complex Metabolic Diseases and Inherited Disorders of the One Carbon Metabolism

**DOI:** 10.3390/cells9081882

**Published:** 2020-08-11

**Authors:** Viola J. Kosgei, David Coelho, Rosa-Maria Guéant-Rodriguez, Jean-Louis Guéant

**Affiliations:** 1UMR Inserm 1256 N-GERE (Nutrition, Génetique et Exposition aux Risques Environmentaux), Université de Lorraine, 54500 Vandoeuvre-lès-Nancy, France; viokos84@yahoo.com (V.J.K.); david.coelho@univ-lorraine.fr (D.C.); rosa-maria.gueant-rodriguez@univ-lorraine.fr (R.-M.G.-R.); 2Departments of Digestive Diseases, Nutrition and Endocrinology and Molecular Medicine and National Center of Inborn Errors of Metabolism, University Hospital Center, Université de Lorraine, 54500 Vandoeuvre-lès-Nancy, France

**Keywords:** Sirtuin1, peroxisome proliferator-activated receptor-γ coactivator-1α, peroxisome proliferator activated receptors, obesity, metabolic syndrome, vitamin B12, folate, fetal programming, inherited metabolic disorders

## Abstract

Sirtuin1 (Sirt1) has a NAD (+) binding domain and modulates the acetylation status of peroxisome proliferator-activated receptor-γ coactivator-1α (PGC1α) and Fork Head Box O1 transcription factor (Foxo1) according to the nutritional status. Sirt1 is decreased in obese patients and increased in weight loss. Its decreased expression explains part of the pathomechanisms of the metabolic syndrome, diabetes mellitus type 2 (DT2), cardiovascular diseases and nonalcoholic liver disease. Sirt1 plays an important role in the differentiation of adipocytes and in insulin signaling regulated by Foxo1 and phosphatidylinositol 3′-kinase (PI3K) signaling. Its overexpression attenuates inflammation and macrophage infiltration induced by a high fat diet. Its decreased expression plays a prominent role in the heart, liver and brain of rat as manifestations of fetal programming produced by deficit in vitamin B12 and folate during pregnancy and lactation through imbalanced methylation/acetylation of PGC1α and altered expression and methylation of nuclear receptors. The decreased expression of Sirt1 produced by impaired cellular availability of vitamin B12 results from endoplasmic reticulum stress through subcellular mislocalization of ELAVL1/HuR protein that shuttles Sirt1 mRNA between the nucleus and cytoplasm. Preclinical and clinical studies of Sirt1 agonists have produced contrasted results in the treatment of the metabolic syndrome. A preclinical study has produced promising results in the treatment of inherited disorders of vitamin B12 metabolism.

## 1. Introduction

Sirtuin1 (Sirt1) is one of the seven mammalian proteins belonging to the silent information regulator 2 (Sir2) proteins/Sirtuin family with highly conserved catalytic and nicotinamide adenine dinucleotide (NAD+) binding domain [1]. Sirtuins (Sirt) catalyze histone and non-histone lysine deacetylation in a NAD+ dependent manner [2,3]. Sirt1 has been shown to play an important role in increasing longevity in a number of lower animals like worms and flies [4,5,6]. Sirt1 is localized in the cytosol and the nucleus, between which it shuttles in response to different pathological and physiological environmental stimuli [2] including cellular stress and metabolic energy dysregulation. Sirt1 regulates cellular energy metabolism through deacetylation of transcription factors and co-factors of energy metabolism [4,7]. Beside its role in metabolic redox electron transfer, NAD+ is an essential Sirt1-dependent cell sensor of energy metabolism, insulin secretion and adaptation to cell stress [8].

### 1.1. Role of Sirt1 on the Regulation of Energy Metabolism by Peroxisome Proliferator-Activated Receptor-Γ Coactivator-1α and Peroxisome Proliferator Activated Receptors

Calorie restriction regulates the expression levels of Sirt1 in a tissue-dependent manner [9]. Sirt1 regulates tissue glucose homeostasis, fatty acid beta oxidation and energy metabolism by modulating the acetylation status of peroxisome proliferator-activated receptor-γ coactivator-1α (PGC1α), Fork Head Box O1 transcription factor (Foxo1) and cAMP response element-binding protein (CREB), according to the nutritional status and fasting state. Sirt1 enhances PGC1α activity by deacetylating its lysine residues [10,11,12]. During fasting, Sirt1 activates PGC1α, which then induces the transcription of genes encoding gluconeogenic enzymes and suppresses the transcription of glycolytic genes in the liver [11]. Deacetylation of Foxo1 by Sirt1 regulates thyroid hormone mediated transcription of gluconeogenic genes like glucose-6-phosphatase and phospho-enoyl pyruvate carboxy kinase in mouse and human hepatic cells [13,14]. Activated Foxo1 binds to insulin response elements (IRE) promoters of these genes to induce their transcription [13,15]. Peroxisome proliferator activated receptors (PPARs) interact with co-regulators, including PGC1α in the regulation of energy homeostasis, insulin sensitivity, inflammation and adipogenesis [16]. Sirt1 regulates lipid metabolism of the liver in response to energy deprivation through deacetylation of PPARα. PPARα target genes encode fatty acid beta oxidation enzymes, carnitine palmitoyltransferase I, medium-chain acyl-CoA dehydrogenase and fatty acyl CoA synthase [17]. Liver specific ablation of Sirt1 induces the decreased transcription of PPARα target genes of fatty acid oxidation [18]. Similarly, the decreased expression of Sirt1 induces dysregulated energy metabolism through imbalanced acetylation and methylation of PGC1α in the myocardium of weaning animals exposed to methyl donor deficiency [19]. Sirt1 inhibits the expression of uncoupling protein gene 2 (UCP2), while Sirt1 silencing produces a mirrored effect in knock out mice [20]. Taken together, these data show that Sirt1 plays a prominent role on the regulation of energy metabolism through the deacetylation of PGC1α and Foxo1.

### 1.2. The Role of Sirt1 in Metabolic Syndrome and Insulin Resistance

The metabolic syndrome is a global public health problem related to overnutrition. It is defined as a cluster of cardio-metabolic abnormalities that includes obesity, insulin resistance or diabetes mellitus type 2 (DT2), hypertension and dyslipidemia [21,22]. The prevalence and incidence of the metabolic syndrome is increasing dramatically worldwide [21,23]. Systemic overexpression of Sirt1 has been reported to have protective effects against physiological damage in mice exposed to a high fat diet [24]. The decreased expression of Sirt1 explains part of the consequences of fatty acid enriched diets on the metabolic syndrome, DT2, cardiovascular diseases, nonalcoholic liver disease [25,26,27,28] and metabolic syndrome-associated cancers [29] (Figure 1). The decreased expression and activity of Sirt1 is involved in the pathogenesis of insulin resistance, DT2 and liver steatosis [30,31]. Decreased Sirt1 expression produces overexpression of miR-180a and impaired insulin signaling in hepatocytes miR-180a targets the 3′ UTR Sirt1 mRNA and is increased in insulin resistant hepatocytes and serum of diabetic patients [32]. Conversely, the overexpression of Sirt1 improves insulin sensitivity of hepatocytes in genetically obese ob/ob mice [33]. Insulin sensitivity is improved through inhibition of endoplasmic reticulum (ER) stress and decreased activity of mammalian/mechanistic target of Rapamycin complex 1 (mTORC1) in these animals [33]. Taken together, these data highlight the decreased activity of Sirt1 as a key component of the molecular mechanisms that produce the outcomes of metabolic syndrome and insulin resistance.

### 1.3. Role of Sirt1 in Pancreatic Beta Cells, Adipose Tissue and Skeletal Muscle

Secretion of insulin by pancreatic beta cells is enhanced by Sirt1 in response to glucose stimulation [34,35,36]. Conversely, decreased expression of Sirt1 impairs glucose-stimulated insulin secretion and expression of mitochondrial genes that control metabolic coupling. Genetic or pharmacologic activation of Sirt1 protects beta cells against lipotoxicity of circulating lipids [37,38]. Furthermore, the specific deletion of Sirt1 in beta cells decreases the expression of glucose transporters and ER chaperones involved in an unfolded protein response [39]. The treatment of human isolated islets with gamma butyric-acid (GABA) enhances Sirt1 activity and attenuates drug-induced apoptosis, suggesting an anti-apoptotic role of Sirt1 [40].

Clinical studies evidenced decreased Sirt1 expression in adipose tissues of obese patients [41,42] and increased expression in progressive weight loss [43,44]. Adipocyte specific ablation of Sirt1 induces increased adiposity and manifestation of metabolic dysfunction including insulin resistance [25]. Overexpression of Sirt1 in adipose tissue in mice enhances glucose homeostasis and prevents age-induced decline in insulin sensitivity [45]. Chronic exposure to a high fat diet accelerates glucose intolerance and hyperinsulinemia in mice with adipocyte specific knockout of Sirt1 [46]. High fat diet induces cleavage of Sirt1 via Caspase 1 activation in adipose tissues [25]. Sirt1 induces browning remodeling of white adipose tissue by deacetylation of PPARγ [47,48]. PPARγ is highly expressed in adipocytes where it acts as a key regulator of lipid metabolism, insulin sensitivity and adipocyte differentiation [41,42,49]. Adiponectin is an adipocyte hormone associated with obesity and type 1 diabetes [50]. Sirt1 regulates the secretion of adiponectin through PPARγ upregulation of ER oxidoreductase α (Erol-Lα) [51]. Sirt1 also upregulates adiponectin secretion by activating Foxo1 and increasing the interaction between Foxo1 and C/EBP α [52].

Sirt1 plays an important role in the differentiation of skeletal muscle [53]. Insulin signaling is regulated by the Foxo1-Sirt1 pathway in skeletal muscle [54]. Sirt1 enhances insulin sensitivity through PI3K signaling in response to caloric restriction [55]. Conversely, Sirt1 silencing decreases insulin sensitivity in Sirt1 KO mice. Wild type mice fed with a caloric restriction diet have a dramatic decreased acetylation of p53 and PGC1α, compared to Sirt1 KO mice [55]. In contrast, the overexpression of Sirt1 in skeletal muscle in vivo does not improve insulin resistance and had little impact on mitochondrial metabolism, suggesting that the overexpression of Sirt1 protein is not the single factor involved in insulin sensitization of skeletal muscle [56]. PPARγ- PGC1α couple is crucial in the regulation of energy metabolism in skeletal muscle. Its inhibition induces insulin resistance in C2C12 skeletal muscle cells, while its overexpression attenuates insulin resistance and enhanced glucose uptake [57]. In summary, Sirt1 acts as a sensor of the nutritional status on molecular mechanisms related with metabolic syndrome and insulin resistance, in pancreatic beta cells, adipose tissue and skeletal muscle.

### 1.4. Anti-Inflammatory Role of Sirt1 and PPAR in Metabolic Syndromes and Related Diseases

Visceral low grade inflammation triggered by adipocytes contributes to the pathogenesis of obesity, insulin resistance and other outcomes of the metabolic syndrome [58,59,60]. Besides dysregulated energy metabolism, increasing evidences link inflammation to pathogenesis of metabolic syndromes and related disorders, including diabetes and obesity [59,60,61]. Inflammation of the pancreatic beta cells in the islets is one of the prominent mechanisms of diabetes type 1 (DT1) and DT2. Adipocyte Sirt1 controls systemic insulin sensitivity through its effects on macrophages of adipose tissue [46,62]. Sirt1 knockdown exhibits elevated expression of Tumor Necrosis Factor α (TNF α) in adipocytes, induces macrophage infiltration and inflammation and increases cytokines levels of interleukin 1 beta (IL-1B), Interleukin 10 (IL-10), Interleukin 4 (IL-4) and TNF α in mice fed with high fat diet [63]. Conversely, overexpression of Sirt1 attenuates the adipose tissue inflammation and macrophage infiltration induced by a high fat diet [63]. Sirt1 inhibits inflammation in adipose tissue via mTOR/p70 ribosomal protein kinase 1 (S6k1) and Akt2 interacting pathways [64].

Converging evidences support an antagonistic crosstalk between Sirt1 and nuclear transcription factor Kappa B (NF-κB) [65]. Damaging effect of proinflammatory cytokines on beta cells is attenuated by the overexpression of Sirt1 in isolated rat islets through the deacetylation of P35 subunit of NF-κB [66]. Sirt1 deacetylates lysine 310 in the RelA/P65 subunit of NF-κB [67,68]. As a consequence, deacetylated RelA/P65 impairs methylation of lysine 314 and 315 residues, leading to ubiquitination and degradation [68,69]. Sirt1 knockdown in 3T3-L1 adipocytes activates the NF-κB signaling pathway by increased acetylation of NF-kB components and impaired interaction with promoters of matrix metalloproteinases and monocyte chemoattractant protein 1 (MCP1) [70].

Mice with hepatocyte specific knock out of Sirt1 (Sirt1LKO) exposed to a high fat diet develop hepatic inflammation and ER stress [18]. Liver inflammation of Sirt1 knock out mice results from increased expression of proinflammatory cytokine including TNF-α and IL-1B and macrophage infiltration. Liver ER stress of Sirt1LKO mice results from increased phosphorylation of translation initiation factor (elf2-α) and C-Jun N-terminal (JNK) [18]. PPARα confers protection against cellular stress and inflammation through various mechanisms related to Sirt1 [71]. PPARα agonist fenofibrate upregulates Sirt1 expression, suppresses CD40 and decreases acetylation of NF-κB-P65 in TNF-α treated 3T3-L1 adipocytes [72]. Chronic LPS stimulation of PPARγ deficient macrophages results in increased production of proinflammatory cytokines and decreased expression of anti-inflammatory cytokine IL-10 [73]. Furthermore, PPARγ deficiency causes delayed monocyte kinetic differentiation into macrophages [73]. In addition, PPARγ plays a crucial role in maturation of alternative phenotype in adipose tissues [74]. In summary, Sirt1 exerts protective effects against cellular stress and inflammation through complementary molecular and cellular mechanisms in adipose tissue, pancreatic islets and liver.

### 1.5. Antioxidant Role of Sirt1 and PPAR against Metabolic Syndromes and Related Diseases

Oxidative stress is associated with pathogenesis of complex metabolic diseases. Sirt1 activates Foxo transcription factors via the feedback loop [75,76]. Activation of Foxo3a and Foxo1a by Sirt1 deacetylation induces the transcription of catalase and manganese superoxide dismutase (MnSOD). Moderate overexpression of Sirt1 is protective against oxidative stress in the mice heart by upregulating the expression of catalase through Foxo1a-dependent mechanisms [77]. In contrast, high levels of Sirt1 increase oxidative stress and cardiomyopathy [77]. A decreased expression of Sirt1, increased expression of NADPH oxidases (P42Phox) and increased superoxide production is observed in monocytes of DT1 patients [78]. The protective role of Sirt1 against oxidative stress is linked to the deacetylation of check point kinase 2, which increases cell death under oxidative stress [79]. We and others have shown a link between Sirt1 and RNA binding proteins like HUR in response to cellular stress [80,81]. Mitochondrion is one of the main organelle involved in ROS production [82]. PGC1α induces expression of superoxide dismutase 2 (SOD2) and glutathione peroxidase involved in ROS detoxification [12,83]. These data illustrate the important role of Sirt1 in oxidative stress homeostasis.

### 1.6. Role of Sirt1 in Fetal Programming and Nutritional and Inherited Disorders of Vitamin B12 Metabolism

The fetal programming hypothesis [84], also named “developmental origins of health and disease hypothesis” (DOHaD), proposes that unfavorable intrauterine life, including intrauterine growth restriction (IUGR), predicts the risk of postnatal complex diseases, including insulin resistance, DT2 and other outcomes of pathological obesity. We have shown that the maternal deficiency in methyl donors (MDD, vitamin B12 and folate) during pregnancy and lactation of rodents produces a low birth weight and an epigenomic Sirt1-dependent dysregulation of mitochondrial energy production and fatty acid oxidation in offspring [85,86]. It is noteworthy that the decreased expression of Sirt1 observed in fetal programming of maternal MDD is also a hallmark of overnutrition and pathological obesity (Figure 2). Moreover, population studies have highlighted an association between maternal methyl donor status and manifestations of fetal programming. In India and Nepal, many babies are thin with central obesity. There is a higher prevalence of mothers with low serum vitamin B12, folate deficiency and intrauterine growth restriction, compared to Europe [87]. In these two countries, the most insulin resistant children are born to mothers who have the lowest serum vitamin B12 at the first trimester of pregnancy [88,89]. A variant of adenosyl-methionine decarboxylase is also associated with childhood obesity, in India [90]. Folate seems to influence the metabolic consequences of fetal programming in Europe, despite contrasted results among population studies [91]. In France, a genetic polymorphism (677C > T, relatively common) of methylenetetrahydrofolate reductase (MTHFR) was associated with low birth weight and high insulin resistance in morbidly obese adolescents [92].

The decreased expression of Sirt1 plays a prominent role in the pathomechanisms of fetal programming produced by MDD in rats (vitamin B12 and folate) [19,81,93,94]. Vitamin B12 is metabolized into two active cofactors, methyl-cobalamin (Me-Cbl) and adenosyl-cobalamin (Ado-Cbl), the cofactors for cytoplasmic methionine synthase (MS/MTR) and mitochondrial methyl malonyl CoA mutase, respectively [95]. Me-Cbl and methyl folate are needed for the transmethylation of homocysteine into endogenous methionine, which is catalyzed by methionine synthase. Methionine is the direct precursor of S-adenosyl methionine (SAM), the universal methyl donor needed for transmethylation reactions involved in epigenomic regulatory mechanisms [86].

MDD during pregnancy and lactation induces impaired fatty acid oxidation, reduced activity of complexes I and II, cardiac hypertrophy with enlargement of cardiomyocyte and liver steatosis in weaning rat pups [19,96]. These observations are linked to the hyperacetylation and hypomethylation of PGC1α and dissociation of PGC1α from PPARα through decreased expression of Sirt1 and protein arginine methyltransferase 1 (PMRT1; Figure 2 and Figure 3) [19]. MDD weakens the activator activity of PGC1α for other nuclear receptors, including estrogen receptor-α (ER-α), estrogen-related receptor-α, hepatocyte nuclear factor 4 (HNF-4) and vitamin D receptor (VDR). The liver steatosis of pups born from mothers with MDD during pregnancy and lactation resulted predominantly from hypomethylation of PGC1α, the decreased binding with its partners, including PPARα and HNF4 and the subsequent impaired mitochondrial fatty acid oxidation [96]. The effects of fetal programming on the liver of rats born from MDD mothers are worsened when pups are subsequently subjected to a high fat diet (HF) after d50 [97]. The MDD/HF animals have hallmarks of steato-hepatitis, with increased markers of inflammation and fibrosis, insulin resistance and key genes triggering the pathomechanisms of non-alcoholic steato-hepatitis (NASH; transforming growth factor beta super family, angiotensin and angiotensin receptor type 1). These data show that MDD during pregnancy is a risk factor of NASH in populations subsequently exposed to a HF diet. The deactivation of PGC1α is also involved in the brain manifestations of MDD fetal programming in rats. MDD during gestation and lactation alters the cerebellum plasticity in offspring, with a lower expression of synapsins. The altered neuroplasticity results from decreased expression and methylation of ER-α and subsequent decreased ER-α/PPAR-γ coactivator 1 α (PGC-1α) interaction in the deficiency condition. The impaired ER-α pathway leads to decreased expression of synapsins through a decreased EGR-1/Zif-268 transcription factor and Src-dependent phosphorylation of synapsins [98]. Deficiencies in methyl donors and in vitamin D are independently associated with altered bone development. In young rats, MDD decreases the total body bone mineral density, reduced tibia length and impaired growth plate maturation, and in preosteoblasts, MDD slows cellular proliferation. MDD produces a decreased expression of VDR, estrogen receptor-α, PGC1α, PRMT1 and Sirt1 and decreased nuclear VDR-PGC1α interaction [99]. The weaker VDR-PGC1α interaction is attributed to the reduced expression and imbalanced methylation/acetylation of PGC1α and the nuclear VDR sequestration by heat shock protein 90 (HSP90). These mechanisms together compromise bone development, as reflected by lowered bone alkaline phosphatase and increased proadipogenic PPARγ, adiponectin and estrogen-related receptor-α expression [99].

The impaired cellular availability of vitamin B12 leads to ER stress related to decreased expression of Sirt1. ER stress is evidenced by increased expression and activation of elf2-α and activating transcription factor 6 (ATF6) and may be favored by decreased expression of heat shock proteins (HSP). Decreased Sirt1 impairs the transcription of HSP by increasing the acetylation of heat shock protein factor 1 (HSF1; Figure 4) [93]. Conversely, overexpression of Sirt1 and HSF1 and activation of Sirt1 by SRT1720 as well as addition of vitamin B12 induce a dramatic decrease of ER stress in NIE115 neuronal cells with impaired cellular B12 availability [93]. Similarly, ER stress is increased in fibroblasts of cblC, cblG and cblG* patients with inherited disorders of cellular vitamin B12 metabolism [100]. Some of the molecular mechanisms that underlie the neurological manifestations of patients with inherited disorders of vitamin B12 metabolism are related to transcriptomic changes of genes involved in RNA metabolism and ER stress. The transcriptomic changes result from the subcellular mislocalization of several RNA binding proteins (RBP), including the ELAVL1/HuR protein implicated in neuronal stress and HnRNPA1 and RBM10, in patient fibroblasts and Cd320 knockout mice with impaired cellular uptake of vitamin B12. The decreased interaction of ELAVL1/HuR with the CRM1/exportin protein of the nuclear pore complex and its subsequent mislocalization result from hypomethylation by decreased SAM and protein methyl transferase CARM1 and dephosphorylation by increased protein phosphatase PP2A. The mislocalization of ELAVL1/HuR triggers the decreased expression of Sirt1 deacetylase and other genes involved in brain development, neuroplasticity, myelin formation and brain aging [81,100]. In summary, Sirt1 plays a prominent role in the pathomechanisms produced by MDD through its role on the acetylation of PGC1α, the transcription of HSP and the subcellular localization of RNA binding proteins (RBP).

### 1.7. Sirt1 Is a Target for the Treatment of Complex and Hereditary Metabolic Diseases

Preclinical and clinical studies in the treatment of complex and inherited metabolic diseases have produced contrasted results in the evaluation of health benefits of activation of Sirt1 by natural and pharmaceutical small activating molecules.

Resveratrol is a natural polyphenolic compound found in red wine and grape and in plants and fruits like berries and peanuts. Resveratrol activates Sirt1 and mimics the caloric restriction status known to be protective against the metabolic syndrome. For decades, the therapeutic use of resveratrol has been considered in regard to its anti-inflammatory, antioxidant, antiaging and anticancer properties [101,102,103]. It reduces oxidative stress, hepatic steatosis and hypertension in high fat diet induced obesity in rodents and non-human primates [104,105,106,107,108]. In addition, resveratrol protects against high fat diet induced hepatic steatosis and endoplasmic stress [104,108,109] by decreasing the expression of proinflammatory cytokines, including TNF- α, IL-1β and IL-6 and increasing antioxidant enzymes like SOD2 and catalase [110]. Treatment of human monocytes in hyperglycemic condition with resveratrol induces upregulated expression of Sirt1, decreased P42Phox expression and upregulates the expression and activation of Foxo3a, which together lead to decreased production of superoxide [78]. Resveratrol has been proposed as a potential therapeutic agent against cardiovascular diseases [107,111,112,113]. It reduces hypertension [114,115] and cardiac hypertrophy via LBK1-AMPK1-eNOS signaling pathways in hypertensive animals [114].

Given the health benefits of resveratrol in experimental animal studies, clinical trials have been carried out to evaluate its effects in the prevention and treatment of metabolic syndrome. A preliminary exploratory trial showed that administration of 500 mg resveratrol twice per day for 60 days in type 1 diabetic patients decreased fasting plasma sugar (FPS) and hemoglobin A1C [116]. A prospective open-label randomized controlled trial reported that a 30 days resveratrol supplementation improves hemoglobin A1c, total cholesterol and systolic blood pressure and insulin sensitivity in patients with diabetes type 2 [117,118]. A meta-analysis of nine randomized clinical trials with 283 participants concluded that resveratrol improves insulin sensitivity (HOMA-IR index) in DT2 patients, and had no effects on hemoglobin A1c and the lipid profile [119]. Similar results were observed in another meta-analysis of 29 randomized clinical trials involving 1069 participants [120]. In contrast, a one month randomized clinical double-blinded crossover administration of 150 mg/day of resveratrol had no effect on peripheral and hepatic insulin sensitivity [121]. This contrasted result could be due to the interaction of metformin and resveratrol in the DT2 recruited patients. Resveratrol has been also evaluated in cardiovascular diseases, neurodegenerative diseases and cancers [113,122,123]. For example, it ameliorates the endothelial dysfunction in diabetic and obese mice through sirtuin 1 and peroxisome proliferator-activated receptor δ (PPARδ) [124].

Activators of Sirt1 such as SRT1720, SRT2183 and SRT1460 are 1000-fold more active than resveratrol. SRT1720 and resveratrol improve insulin sensitivity in nutritionally and genetically induced obesity and DT2 in mice [125]. SRT1720 treatment improved insulin sensitivity of liver, skeletal muscle and adipose tissue in insulin resistance Zucker *fa*/*fa* rats [126]. SRT1720 enhances fatty acid oxidation through direct deacetylation of PGC1α, Foxo1 and indirect activation of the AMP activated protein kinase (AMPK) pathway [126]. Furthermore, SRT1720 extends the lifespan of mice fed a high fat diet, decreases hepatic steatosis, increases insulin sensitivity and reverses inflammation and oxidative stress markers in adult obese mice subjected to a high fat diet [127]. SRT1720 impairs lipopolysaccharide stimulated inflammatory pathways and TNF-α secretion in macrophages and adipose tissues of Zucker fatty rats [70]. SRT1720 also activates AMPK in a Sirt1-independent manner. However, SRT1720 cannot be used in humans because of potential toxicity. Metformin regulates gluconeogenesis in *ob*/*ob* mice by upregulating hepatic Sirt1 and GCN5 [128]. A computational study confirmed that metformin directly activates Sirt1 [129]. At low NAD+ concentration, leucine and low dose of metformin synergistically activate Sirt1 and AMPK, with enhanced energy metabolism and insulin sensitivity in muscle cells and hepatic cells in vitro [130]. Recently two novel Sirt1 activators with high affinity for Sirt1, SC1C2 and SC1C2.1 attenuated doxorubicin induced DNA damage via deacetylation of P53 and cellular senescence in HepG2 and H9c2 cell lines [131].

Phase 1 clinical trials showed that SRT2104, another small molecule activator of Sirt1 is safe and tolerable in healthy volunteers, including the elderly [132,133]. However, in a prospective double blinded random placebo control crossover study, the daily administration of 2.0 g of SRT2104 for 28 days had a neutral cardiovascular effect in DT2 patients, even if it induced a loss in body weight [134]. Lipid profile of healthy smokers was improved after 28 days SRT2104 administration but again the cardiovascular effects were neutral [134,135].

Some severe forms of inherited disorders of intracellular metabolism of vitamin B12 are resistant to conventional treatments. Decreased Sirt1 activity plays a central role in some of the pathomechanisms of these disorders. We therefore evaluated the effect of Sirt1 agonists in a preclinical study in fibroblasts from patients with cblG and cblC inherited defects of vitamin B12 metabolism and an original transgenic mouse model of methionine synthase deficiency specific to neuronal cells. Patient fibroblasts with cblC and cblG defects of vitamin B12 metabolism presented with endoplasmic reticulum stress, altered subcellular localization of HuR, HnRNPA1 and RBM10, global mRNA mislocalization and increased HnRNPA1-dependent skipping of interferon regulatory factor 3 (IRF3) exons. SRT1720 inhibited ER stress and rescued RBP and mRNA mislocalization and IRF3 splicing. Furthermore, Sirt1 activation by SRT1720 partially restored the methylation and phosphorylation of these RBPs in patients’ fibroblasts. Interestingly, SRT1720, vitamin B12 and SAM treatment improved cognitive functions in conditional *MTR*-KO mice with brain specific invalidation of *Mtr* gene encoding MS enzyme [100]. In particular, SRT1720 improved the deficient hippocampo-dependent learning ability of the mice.

In summary, preclinical and clinical studies of Sirt1 agonists have produced contrasted results in the treatment of the metabolic syndrome. A preclinical study has produced promising results in the treatment of inherited disorders of vitamin B12 metabolism.

## 2. Conclusions

Numerous experimental evidence show a strong link between decreased Sirt1 and the pathological manifestations of metabolic syndrome by synergistic cellular and molecular mechanisms. It is noteworthy that over nutrition and MDD both produce a decrease of Sirt1. There are additive and synergistic effects of the MDD fetal programming and subsequent exposure to a high fat diet in adult life. The therapeutic prospects for using activators of Sirt1 in the treatment of disease outcomes of metabolic syndrome have not been conclusive to date. However, pharmacological activation of Sirt1 opens promising perspectives for the treatment of rare diseases of vitamin B12 metabolism with particular effects on reticulum stress and mislocalization of RBPs.

## Figures and Tables

**Figure 1 cells-09-01882-f001:**
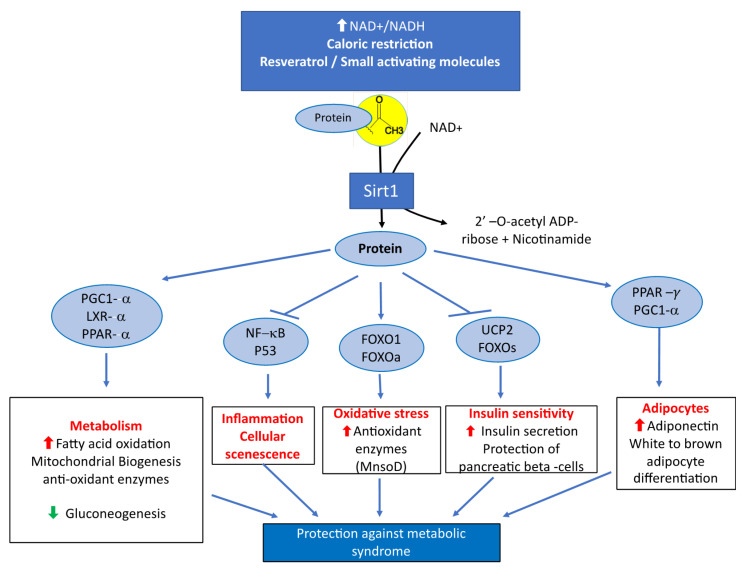
Main metabolic effects of Sirtuin1 related to its protective influence against the manifestations of metabolic syndrome.

**Figure 2 cells-09-01882-f002:**
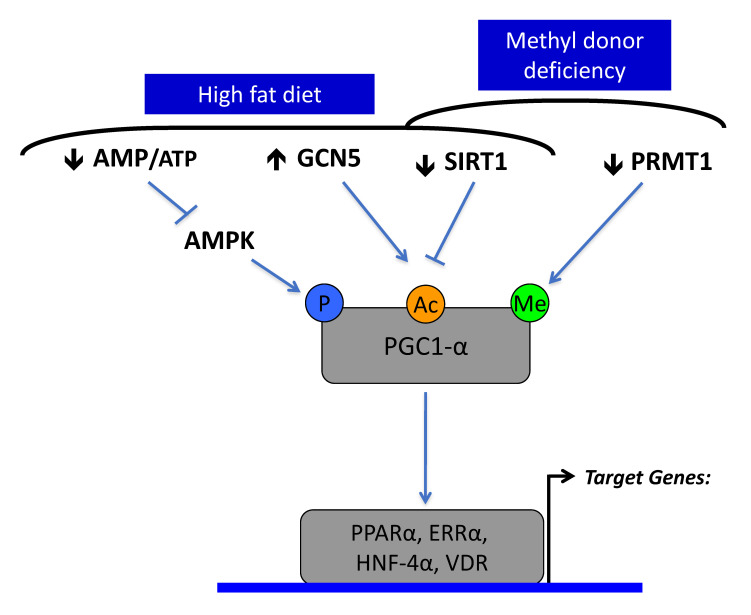
The decreased expression and activity of Sirtuin1 is a common molecular hallmark in a high fat diet and methyl donor deficiency (MDD). Sirtuin1 targets the acetylation of peroxisome proliferator-activated receptor-γ coactivator-1α (PGC1α) and has a complementary role with GCN5 acetylase, AMP kinase and protein arginine methyltransferase (PRMT1) in the regulation of PGC1α activation of nuclear receptors, including PPARα, ERRα, HNF-4α and VDR. PGC1α is phosphorylated and acetylated under the control of AMP kinase (AMPK), GCN5 acetylase and SIRT1 deacetylase. High fat diet and over nutrition decrease activity of AMP kinase, through high intracellular ATP levels, leading to decreased phosphorylation of PGC-1 α. It produces hyperacetylation of PGC-1α through increased expression of GCN5 and decreased expression and activity of SIRT1. MDD produces similar effects through decreased SIRT1 and PRMT1.

**Figure 3 cells-09-01882-f003:**
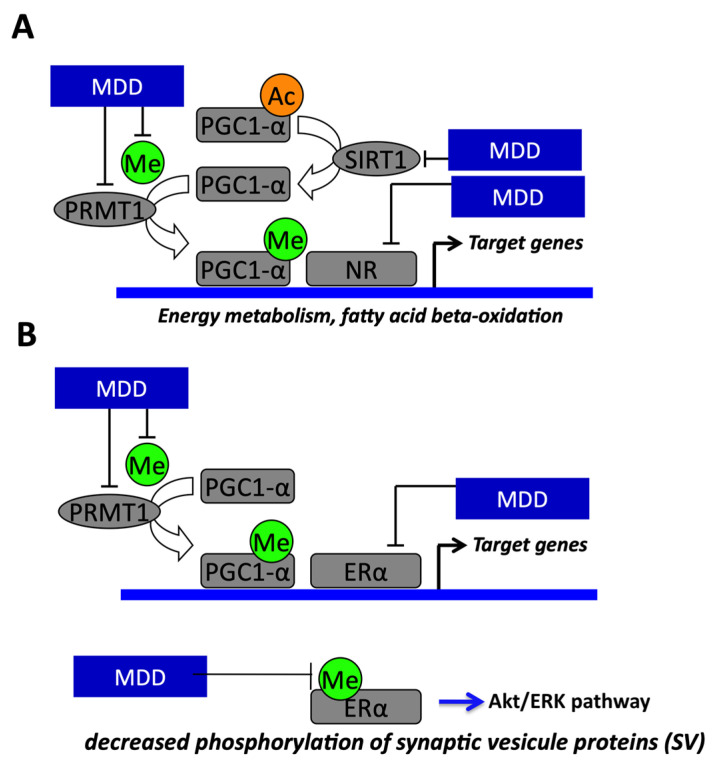
Molecular mechanisms demonstrating the link between methyl donor deficiency (MDD) and fetal programming and energy metabolism in liver and heart and regulation of synapsin expression in neuron. (**A**) The effects of fetal programming on the heart, liver and brain of rats born from methyl donor deficient (MDD) mothers are related to impaired PGC-1α activity through decreased expression of Sirtuin1 (Sirt1) and protein arginine methyltransferase (PRMT1) and decreased synthesis of the universal methyl donor (Me) methyl S- adenosyl methionine (SAM). The decreased activity of PGC-1α results from imbalanced methylation and acetylation. PGC-1α is a regulator of lipid metabolism and fatty acid oxidation through its role as coactivator of PPARα in heart and liver. (**B**) MDD induces decreased phosphorylation of synaptic vesicle proteins through impaired ERα activity linked to decreased SAM levels, PMRT1 expression and PGC-1α activity. PGC-1α is a regulator of synapsin expression and vesicle transport through its role as a coactivator of ERα in the brain.

**Figure 4 cells-09-01882-f004:**
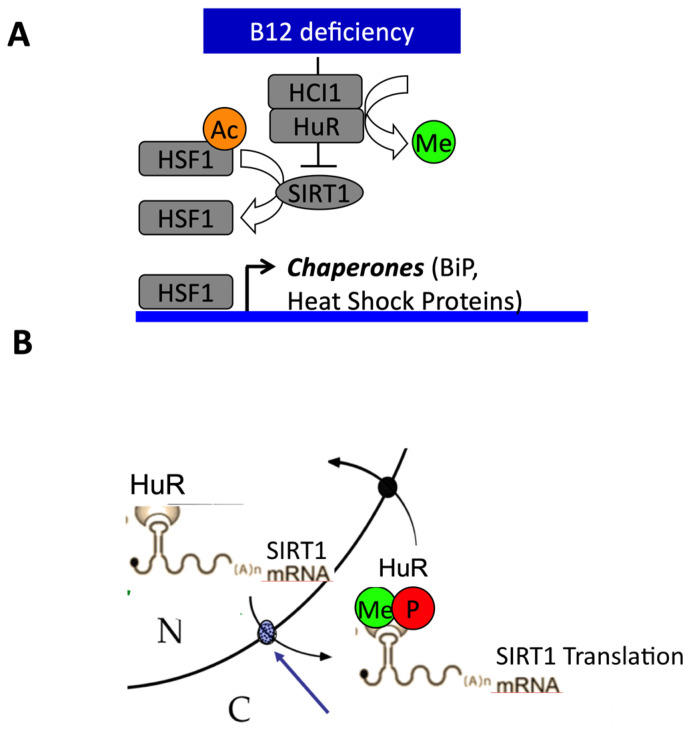
Influence of vitamin B12 cellular availability on endoplasmic reticulum (ER) stress-related decreased expression of Sirtuin1 (Sirt1). (**A**) The decreased cellular availability in B12 activates ER stress pathways and decreases the expression of heat shock proteins through decreased expression of Sirt1 and subsequent hyperacetylation of heat shock factor 1 (HSF1). (**B**) The decreased cellular availability in B12 produces the subcellular mislocalization of the ELAVL1/HuR RNA binding protein implicated in response to ER stress through hypomethylation by decreased synthesis of methyl S-adenosyl methionine (SAM) and dephosphorylation by increased protein phosphatase PP2A. The blue arrow shows the nuclear pore complex. The mislocalization of ELAVL1/HuR triggered the decreased expression of Sirt1 by altered Sirt1 mRNA export from nucleus to cytoplasm.

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
