# Peer review of "Sirt1-PPARS Cross-Talk in Complex Metabolic Diseases and Inherited Disorders of the One Carbon Metabolism"

_cells, 2020, doi:10.3390/cells9081882_

Round 1

Reviewer 1 Report

The manuscript presented by Kosgel and co-authors is a review of the role of SIRT1 in complex metabolic diseases and inherited disorders. After introducing the role of SIRT1, author’s present examples of pathologies in which sirtuin1 has been involved. The main disorder which is presented is the metabolic syndrome and insulin resistance, with the role of Sirt1 in different tissues, in oxidative stress and inflammation. The second important aspect of the review is the role of sirtuin 1 in fetal programming and nutritional inherited disorders. Finally, the authors present a review of the experimental and clinical trials that have been attempted with sirtuin 1 natural or synthetic activators to treat in particular metabolic syndrome and insulin resistance.

1. In general the review presents a rather exhaustive report of the current literature on the subject. It is well organized and well illustrated. The main criticism is that some chapters, after a rapid introduction are a list of results from the literature but without any conclusion. It appears thus necessary to add a conclusion of the different articles reported, at the end of each paragraph.

2. I would also recommend a careful editing of the manuscript as in many places, words are missing or some sentences are long and difficult to understand (for example page 9 l. 253-253).

Few examples:

p. 4 l. 150: “Activation Foxo3a and Foxo1a…..” Activation of or activating?

Legend of Figure 2 seems incomplete.

p.7, l. 191 starts by ”HNF4α, VDR.” Also here the sentence is incomplete.

p. 7, l. 207: These observations

p. 5, l. 158: “Mitochondrial is one of the main organelle…” should be “Mitochondria are one of the main organelles…”

3. All abbreviations should be described in full at the first mention (for example: NASH is not defined). Since there are a lot of abbreviations in the text, a list of these would be helpful.

4. p. 10, l. 314-315: You say that SRT1720 cannot be used in humans. Could you explain why?

Author Response

Reviewer #1

  1. “In general the review presents a rather exhaustive report of the current literature on the subject. It is well organized and well illustrated. The main criticism is that some chapters, after a rapid introduction are a list of results from the literature but without any conclusion. It appears thus necessary to add a conclusion of the different articles reported, at the end of each paragraph.”

We have added these sentences at the end of each paragraph:

“Taken together, these data show that Sirt1 plays a prominent role on the regulation of energy metabolism through the deacetylation of PGC1α and Foxo1.”

“Taken together, these data highlight the decreased activity of Sirt1 as a key component of the molecular mechanisms that produce the outcomes of metabolic syndrome and insulin resistance.”

“In summary, Sirt1 acts as a sensor of the nutritional status on molecular mechanisms related with metabolic syndrome and insulin resistance, in pancreatic beta cells, adipose tissue and skeletal muscle.”

“In summary, Sirt1 exerts protective effects against cellular stress and inflammation through complementary molecular and cellular mechanisms in adipose tissue, pancreatic islets and liver.”

“These data illustrate the important role of Sirt1 in oxidative stress homeostasis.”

“In summary, Sirt1 plays a prominent role in the pathomechanisms produced by MDD through its role on the acetylation of PGC1α, the transcription of HSP and the subcellular localization of RNA binding proteins (RBP).”

“In summary, preclinical and clinical studies of Sirt1 agonists have produced contrasted results in the treatment of metabolic syndrome. A preclinical study has produced promising results in the treatment of inherited disorders of vitamin B12 metabolism.”

  1. “I would also recommend a careful editing of the manuscript as in many places, words are missing or some sentences are long and difficult to understand (for example page 9 l. 253-253).”

We have made the editing throughout the revised version.

  1. All abbreviations should be described in full at the first mention (for example: NASH is not defined). Since there are a lot of abbreviations in the text, a list of these would be helpful.

We have indicated the meaning of NASH and we have added a list a abbreviations.

  1. “p. 10, l. 314-315: You say that SRT1720 cannot be used in humans. Could you explain why?”

We have indicated that the toxicity of SRT1720 limits its use in human

Reviewer 2 Report

Sirtuin1 (Sirt1) regulates lipid metabolism of the liver in response to energy deprivation through deacetylation of PPARα and PPARα target genes. This review article summarized the recent reports regarding that Sirt1 agonists have produced contrasted results in the treatment of metabolic syndrome. I like to give the following comments.

  1. This submission is belonged to the special issue of PPARs. It seems better to add the relationship with PPARs in title.
  2. In the abstract, DT2 must show in clear at the first time.
  3. The pharmacological activation of Sirt1 opens promising perspectives for the treatment of rare diseases of vitamin B12 metabolism that needs the reference(s).
  4. PPARdelta seems not associated with the influence of Sirt1 in this report. Why?
  5. SRT1720 activates AMPK in a Sirt1-independent manner. Is it associated with improvement in the deficient hippocampal-dependent learning ability?
  6. Resveratrol has widely been studied in basic research. It seems to apply in clinical practice. Why?
  7. The inherited vitamin B12 metabolism disorders are resistant and associated with the decreased Sirt1 activity. Role of PPARs in this change is unknown.
  8. “Perspective” section may assist this report to be clear in the future.

Author Response

Reviewer 2

  1. “This submission is belonged to the special issue of PPARs. It seems better to add the relationship with PPARs in title.”

We have modified the title “SIRT1-PPARs cross-talk in complex metabolic diseases and inherited disorders of the one carbon metabolism”

  1. “In the abstract, DT2 must show in clear at the first time.”

This is now indicated

3.”The pharmacological activation of Sirt1 opens promising perspectives for the treatment of rare diseases of vitamin B12 metabolism that needs the reference(s).”

We added cited the reference [100]

  1. “PPARdelta seems not associated with the influence of Sirt1 in this report. Why?”

Few data have been published regarding this question. One article has been published on the effect of resveratrol on endothelial cells dysfunction through a cross-talk between PPARδ and SIRT1. We have added the reference as ref 124 in the revised manuscript.

  1. “SRT1720 activates AMPK in a Sirt1-independent manner. Is it associated with improvement in the deficient hippocampal-dependent learning ability?”

This is an excellent suggestion which has not been studied up to now

  1. “Resveratrol has widely been studied in basic research. It seems to apply in clinical practice. Why?”

 Resveratrol has wide effects that are not restricted to its agonist properties on SIRT1 activity. These effects are listed in the paragraph “Sirt1 is a target for treatment of complex and hereditary metabolic diseases”
